# Repurposing eflornithine to treat a patient with a rare ODC1 gain-of-function variant disease

Surender Rajasekaran[1,2,3†]*, Caleb P Bupp[3,4†], Mara Leimanis-Laurens [1,3], Ankit Shukla[5], Christopher Russell[2], Joseph Junewick[6], Emily Gleason[2], Elizabeth A VanSickle[4], Yvonne Edgerly[2], Bryan M Wittmann[7], Jeremy W Prokop[3,8], André S Bachmann[3]*

[1]Pediatric Critical Care Medicine, Helen DeVos Children's Hospital, Grand Rapids, United States; [2]Spectrum Health Office of Research and Education, Grand Rapids, United States; [3]Department of Pediatrics and Human Development, College of Human Medicine, Michigan State University, Grand Rapids, United States; [4]Medical Genetics, Spectrum Health and Helen DeVos Children's Hospital, Grand Rapids, United States; [5]Department of Pharmacy, Helen DeVos Children's Hospital, Grand Rapids, United States; [6]Department of Diagnostic Radiology, Spectrum Health and Helen DeVos Children's Hospital, Grand Rapids, United States; [7]Metabolon, Morrisville, United States; [8]Department of Pharmacology and Toxicology, College of Human Medicine, Michigan State University, East Lansing, United States

*For correspondence:
surender.rajasekaran@
helendevoschildrens.org (SR);
bachma26@msu.edu (ASB)

†These authors contributed equally to this work

## Abstract

**Background:** Polyamine levels are intricately controlled by biosynthetic, catabolic enzymes and antizymes. The complexity suggests that minute alterations in levels lead to profound abnormalities. We described the therapeutic course for a rare syndrome diagnosed by whole exome sequencing caused by gain-of-function variants in the C-terminus of ornithine decarboxylase (ODC), characterized by neurological deficits and alopecia.

**Methods:** *N*-acetylputrescine levels with other metabolites were measured using ultra-performance liquid chromatography paired with mass spectrometry and Z-scores established against a reference cohort of 866 children.

**Results:** From previous studies and metabolic profiles, eflornithine was identified as potentially beneficial with therapy initiated on FDA approval. Eflornithine normalized polyamine levels without disrupting other pathways. She demonstrated remarkable improvement in both neurological symptoms and cortical architecture. She gained fine motor skills with the capacity to feed herself and sit with support.

**Conclusions:** This work highlights the strategy of repurposing drugs to treat a rare disease.

**Funding:** No external funding was received for this work.

## Introduction

Ornithine decarboxylase (ODC) is a rate-limiting enzyme in the biosynthesis of polyamines (putrescine, spermidine, spermine), which orchestrate essential physiological and pathologic processes including embryogenesis, organogenesis, and neoplastic cell growth (*Bello-Fernandez et al., 1993*; *Pendeville et al., 2001*). We recently described a new autosomal dominant genetic disorder (Bachmann-Bupp syndrome, OMIM #619075) caused by a heterozygous de novo variant in the *ODC1* gene in a 3-year-old girl with phenotypic features that included alopecia universalis and global developmental delay (*Figure 1*; *Bupp et al., 2018*). The nonsense variant caused premature abrogation

**Figure 1.** Patient phenotypes and metabolites before and after eflornithine treatment. Panel **A** shows the timeline of events for the patient with milestones marked on the top and clinical observations below. Panels **B-C** show hair growth and muscle tone are the most noticeable phenotype changes with treatment. Follicular cysts recurred on back, neck, and posterior scalp (bottom left images). First hair growth was eyebrows 1 month into treatment (bottom right images). Panel **D** shows MRI before and after eflornithine treatment. Neonatal: Axial T1 (TR 483 ms, TE 9 ms, and flip angle 63 degrees), T2 (TR 3250 ms, TE 220 ms, and flip angle 90 degrees), and T2-FLAIR (TR 8002 ms, TE 122 ms, and flip angle 90 degrees) show marked abnormal signal of cerebral white matter (*) and several subependymal cysts (arrows). Five years of age: Axial T1 (TR 809 ms, TE 16 ms, and flip angle 111 degrees), T2 (TR 4850 ms, TE 107 ms, and flip angle 142 degrees), and T2-FLAIR (TR 6002 ms, TE 91 ms, and flip angle 90 degrees) show decrease in cerebral white matter volume, but normalization of signal and resolution of subependymal cysts.

of 14-aa residues in the C-terminus of the protein (ODC, p·K448X, *Figure 2*), leading to enhanced function. Red blood cells from the patient exhibited elevated ODC activity and putrescine levels compared to healthy controls. Four additional patients with similar mutations and phenotypic features of this syndrome have since been reported (*Rodan et al., 2018*) and at least four more cases have been identified.

Remarkably, these patients all represent human phenotypes of a transgenic mouse described in 1996, overexpressing C-terminally deleted ODC in the dermal tissue, leading to higher ODC enzyme activity and increased putrescine biosynthesis (*Soler et al., 1996*). The phenotypic changes first described in a mouse model included hair loss that was reversible with ODC inhibitor α-difluoromethylornithine (DFMO; common name eflornithine) (*Soler et al., 1996*). Experiments with the patient's cultured primary dermal fibroblasts showed eflornithine reduced ODC activity, resulting in

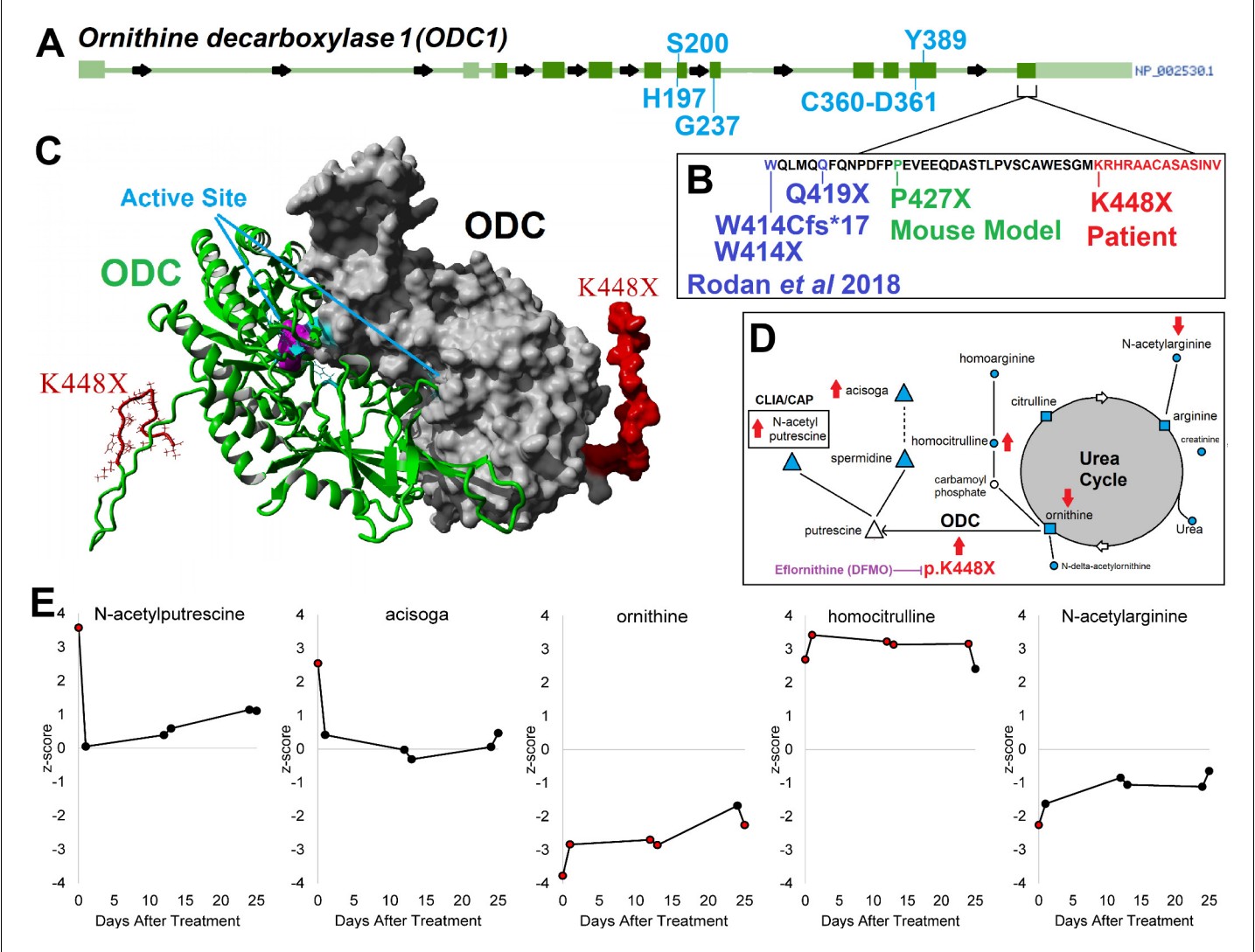

**Figure 2.** *ODC1/ODC clinical variant c.1342 A > T/ p.K448X* and support for eflornithine treatment. Panel **A** shows the gene structure for ODC1 (ornithine decarboxylase 1) with active site amino acids labeled in blue and the last exon identified. In the last exon cluster, a mouse model variant and four different patient variants including our patient's K448X variant are described (Panel **B**, red). The patient variant falls on the disordered C-terminus of ODC, where the two active sites are composed of amino acids from each of two ODC proteins forming a dimer (Panel **C**). The patient with K448X variant displays alterations of metabolic pathways (Panel **D**) including polyamines (triangle), urea (square), and others (circle). Metabolites measured are marked in cyan and those altered by K448X with red arrows based on direction of changes seen in the patient. Panel **E** shows changes in metabolite levels during treatment with eflornithine, with elevated levels of N-acetylputrescine and acisoga decreasing on therapy.

putrescine levels comparable to controls without affecting cell morphology or inducing cell death (*Schultz et al., 2019*).

Based on previously published murine data with eflornithine for gain-of-function variants (*Soler et al., 1996*), multiple long-term safety studies for clinical use in African sleeping sickness (trypanosomiasis), colorectal cancer, and neuroblastoma (*Alirol et al., 2013*; *Priotto et al., 2009*; *Saulnier Sholler et al., 2015*; *Meyskens et al., 2008*), and the absense of toxicity in the patient's primary cell culture inresponse to eflornithine (*Schultz et al., 2019*), we surmised eflornithine might be a novel therapy for patients with this syndrome.

# Materials and methods

**Key resources table**

| Reagent type (species) or resource | Designation | Source or reference | Identifiers | Additional information |
|---|---|---|---|---|
| Gene (*Homo sapiens*) | *ODC1* | NCBI Gene | Gene ID: 4953 | https://www.ncbi. nlm.nih.gov/gene/4953 |
| Chemical compound, drug | *Eflornithine (DFMO)* | Sanofi Aventis | Supplied for study | https://pubchem.ncbi. nlm.nih.gov/compound/ Eflornithine |
| Biological sample (*Homo sapiens*) | Blood EDTA tubes | | | Freshly isolated blood from patient |
| Software, algorithm | YASARA | YASARA | http://www. yasara.org/ | Protein modelling |
| Commercial assay or kit | *Liquid chromatography paired mass spectrometry* | Metabolon, Morrisville, NC | https://www. metabolon.com/ | |

## Study participants and consent

Following FDA approval of our single-patient Investigational New Drug (IND) Application (144022) as a compassionate use treatment protocol, the study was further reviewed and approved by the Spectrum Health Institutional Review Board (IRB). IRB approval for sample collection with informed consent was received to conduct global metabolomics that included, among others, the polyamine metabolites spermidine and *N*-acetylputrescine.

The patient first presented at Spectrum Health, Helen DeVos Children's Hospital (Grand Rapids, MI) at 11 months of age (*Figure 1A*). We diagnosed the ODC C-terminal deletion at age 19 months through whole exome sequencing and characterized the metabolic dyshomeostasis by 32 months (ODC protein and polyamine abnormalities). Eflornithine oral solution was prepared by diluting the lyophilized powder with purified water to a final concentration of 100 mg/mL. At age 4 years and 8 months, we started eflornithine (Sanofi Aventis) treatment with 500 mg/m$^2$/dose bid twice daily via a gastrostomy tube along with a low polyamine diet on November 14, 2019, for 3 months, increasing to 750 mg/m$^2$/dose twice daily, and a final increase to 1000 mg/m$^2$/dose twice daily after 3.5 months. Dosing was based on what had been demonstrated to be safe in pediatric patients in maintenance therapy for neuroblastoma treated with eflornithine (*Saulnier Sholler et al., 2015*).

## Blood collection and processing

EDTA blood tubes collected from the patient were mixed by inversion 8–10 times, centrifuged at $1000 \times g$ for 10 min at 4°C to separate plasma (minimum of 0.25 mL, free of hemolysis from red blood cells) from cellular fraction, and both fractions were immediately placed at −80°C. The plasma specimens were coded and anonymized, kept frozen, and shipped in batch to Metabolon, Morrisville, NC, for metabolomics analysis. *N*-acetylputrescine, the only polyamine that meets CAP/CLIA standards in this analysis, served as the primary indicator of putrescine levels. *N*-acetylputrescine and additional metabolites meeting CAP/CLIA standards (*Figure 2E*) and supplemental metabolites were measured in the EDTA plasma samples using ultra-performance liquid chromatography paired with mass spectrometry and Z-scores were calculated for each metabolite against a reference cohort of 866 pediatric patients as described previously (*Squitti et al., 2019*).

Blood draws for polyamine levels were obtained at initiation of therapy, 1-week post-initiation, immediately prior to each dose increase and then 7 days after each dose increase. These draws were performed in conjunction with safety screening, which included a complete blood count, liver function test, lactate dehydrogenase, complete metabolic panel, calcium, magnesium, and phosphate.

## Results

The novel treatment of this ultra-rare (less than 10 known cases) genetic syndrome presented unique challenges for monitoring efficacy over time. Growth parameters and metabolite levels were monitored easily, whereas others such as cognitive and motor functioning proved challenging, making us dependent on her standardized neurological examination.

### Eflornithine improves clinical findings

The patient was born with a full head of silver-blond hair similar to a previously described murine phenotype (*Soler et al., 1996*), which fell out in early months and she remained hairless other than a few scattered, long hairs on the scalp. One month into treatment, hair growth was noted, with eyebrows appearing first (*Figure 1B–C*). Two months into treatment, scalp hair began to diffusely appear in the normal pattern of hair distribution increasing to resemble normal growth for age (*Figure 1*). She had a history of recurring follicular cyst formation and enlargement. Multiple lesions located on the posterior scalp and back that first were small maculopapular pustules slowly increased in size to approximately 6–7 cm in diameter (*Figure 1B*). These lysed spontaneously, but some would enlarge until painful, requiring surgical removal. Upon initiation of eflornithine, the formation of cysts ceased immediately (*Figure 1B*).

Prior to therapy, she had delayed development which manifested with no standing, cruising, or sitting, and limited fine motor skills. Her BMI increased during eflornithine treatment from 25th percentile to 90th percentile primarily due to increase in weight. This quantitative change was not accompanied by any change in body habitus but rather an increase in muscle bulk. She gained muscle strength demonstrated by acquisition of her ability to hold up her head without support (*Figure 1B*). As the video file shows after 4 months of eflornithine therapy, she was able to sit unsupported and maintain posture with the physical therapist providing resistance, use a walker, and feed herself with a spoon with some assistance. *Video 1* allows for optimal visualization of this rapid improvement of our patient with this gain-of-function mutation in the *ODC1* gene. The drastic external change in hair growth, and visible improvement in coordination, attention, and interaction can be clearly seen.

A neonatal brain MRI showed abnormal cerebral white matter and subependymal cysts (*Figure 1D*). Repeat MRI done at the end of the 9-month treatment trial with eflornithine demonstrated normalization of the cerebral white matter signal with decrease in volume with white matter loss and resolution of all previously noted cysts. Post-treatment magnetic resonance spectroscopy was also performed showing normalization of the *N*-acetylaspartate and choline signals relative to creatine *Figure 1D*.

### Eflornithine normalizes metabolomic findings

*N*-acetylputrescine, the only polyamine metabolite measurement that is CAP/CLIA-certified, was quantified in addition to others using a global metabolomics approach (*Figure 2*) before and after initiating therapy with eflornithine. Metabolite levels from a reference cohort of 866 pediatric patients were converted into Z-scores, a calculation of standard deviations from the mean of the reference populationthat our patient's values are compared to. In the global analysis of 915 metabolites of the patient before treatment, a total of 16 had values above the 97.5th percentile and 38 below the 2.5th percentile, with a noted difference in polyamine connected metabolites (*Source data 1*) without any marked disruption of any other metabolic pathways on treatment. The initial elevation of both *N*-acetylputrescine as well as the polyamine metabolite *N*-(3-acetamidopropyl) pyrrolidin-2-one (acisoga), which were above the 97.5th percentile, decreased at initiation of therapy and remained reduced at all time points (*Figure 2*), indicating that eflornithine treatment had the expected effect. Ornithine and *N*-acetylarginine were below the 2.5th percentile at start of therapy and normalized to the larger pediatric values over the course of therapy. Urea cycle components citrulline and arginine, along with other metabolites, remained at 1 to −1 standard deviation throughout the study period (*Figure 2*).

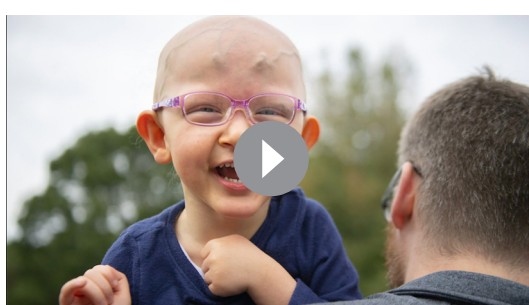

**Video 1.** Treatment progression after 4 months of eflornithine therapy.
https://elifesciences.org/articles/67097#video1

## Discussion

The introduction of both whole genome and exome sequencing into clinical practice has led to rare diseases being diagnosed at rates never before seen (*Turro et al., 2020*; *Tarailo-Graovac et al., 2016*; *Splinter et al., 2018*). There are over 6000 rare diseases (incidence of greater than 1 in 2000 people) with over 300 million people worldwide affected (*Nguengang Wakap et al., 2020*). Though collectively common, each rare disease is unique making it challenging to develop specific therapies.

The process of developing treatment options for these rare diseases starts with a description of the genetic abnormality and developing an understanding of the molecular disruptions downstream from the affected protein. Once the biochemical perturbations are identified, then the quest to identify a drug that will return molecular pathways to normal begins. In genetic diseases, the correct mechanism to adopt could be challenging as many options exist such as activating or repressing pathways, enzyme blockade therapy, gene therapy regimens (*Mendell et al., 2017*), and potentially circumventing or correcting a genetic mutation such as treatments for muscular dystrophy (*Iyer et al., 2019*). Taking a genetic defect to human trials requires cell cultures, animal studies, and phased trials to determine safety and efficacy of such therapies. For multiple patients, often with unique genetic variants, to see benefit from this process could take the best part of a decade even as identification of successful drugs has been enhanced by the Orphan Drug Act (*Augustine et al., 2013*; *Griggs et al., 2009*).

We show a more rapid strategy of matching a patient with an ultra-rare newly identified syndrome to a drug with subsequent treatment being able to safely correct many phenotypic features. Once the whole exome identified the biochemical pathway, we used data from a previously described transgenic mouse model and our published cell culture study to surmise, eflornithine therapy could be of benefit to the patient. Though experimental data suggested that eflornithine could be beneficial, there is a chasm between ex vivo and in vivo studies with difficulties in extrapolating efficacy or safety from a fibroblast study alone (*Schultz et al., 2019*). We were fortunate that studies existed for eflornithine in a large enough population to suggest dosage and safety (*Alirol et al., 2013*; *Priotto et al., 2009*; *Saulnier Sholler et al., 2015*; *Meyskens et al., 2008*).

Once therapy was initiated, some neurological improvement in the patient was noted with better posture, weight gain, reduction, elimination of cyst formation, and significant hair growth. Six months into therapy, she had fine motor capability that she previously lacked, such as the capacity to feed herself and sit with some support. Brain imaging also showed changes that are beyond what would be explained merely by the passage of time, suggesting improvement related to eflornithine treatment.

While COVID-19 restrictions interrupted neurological assessments over the treatment period, the improvements noted throughout the relatively short treatment period of 6 months are truly remarkable, especially given the neurological deterioration in the patient prior to eflornithine therapy. This could be especially consequential if we could initiate therapy in a neonate diagnosed early before neurological damage occurs. We are now aware of other patients identified that present with similar gain-of-function ODC variants and polyamine abnormalities such as elevated *N*-acetylputrescine (*Rodan et al., 2018*). The therapy outlined here should allow for replication of the findings with a promise for significant improvement in quality of life for these patients. For such patients we recommend continued monitoring of multiple metabolites including *N*-acetylputrescine and acisoga to ensure that eflornithine dosing and urea/polyamine metabolite levels stay within normal ranges. The advent of global metabolomics presents a unique opportunity not only to develop a complete understanding of the dyshomeostasis prior to therapy but also a way to appreciate the drug's impact on interconnected metabolic cycles simultaneously and perhaps a means of identifying disruptions early and predicting adverse effects. This may lead to earlier initiation of therapy in future patients,

thereby perhaps avoiding some of the neurological delay that has come to characterize the disease in our patient.

## Conclusion

In this study we have laid forth a promising example of going from first publication of a new syndrome to FDA-approved single-patient investigational repurposed drug treatment in 16 months, a methodology and speed rarely seen in the clinical science of rare diseases.

# Acknowledgements

The authors wish to thank the patient and family for their participation. We would like to dedicate this article to our patient who is a first in so many ways, and to her incredible family. We acknowledge the support we received from the research team at Spectrum Health. We are most grateful to Sanofi-Aventis for providing eflornithine for this study. We also thank Dr B Keith English, MD, Charles Schwartz, PhD, and Brittany Essenmacher for the critical review and editing of this manuscript, David Tack, PhD, for the design of some of the figures, and Olivia Verburg for her help with processing samples and logging samples.

# Additional information

### Competing interests

Surender Rajasekaran, Caleb P Bupp, André S Bachmann: Inventor on a pending US patent (US-2020215010-A1), methods for treating or preventing developmental disorders associated with mutations in the OCD1 gene. The other authors declare that no competing interests exist.

### Funding

No external funding was received for this work.

### Author contributions

Surender Rajasekaran, Conceptualization, Investigation, Project administration; Caleb P Bupp, Conceptualization, Investigation; Mara Leimanis-Laurens, Investigation, Methodology; Ankit Shukla, Methodology; Christopher Russell, Resources, Supervision; Joseph Junewick, Elizabeth A VanSickle, Writing - review and editing; Emily Gleason, Resources; Yvonne Edgerly, Supervision; Bryan M Wittmann, Formal analysis, Methodology; Jeremy W Prokop, Data curation, Investigation, Visualization; André S Bachmann, Conceptualization, Methodology, Writing - original draft, Writing - review and editing

### Author ORCIDs

Surender Rajasekaran (ID) https://orcid.org/0000-0002-4430-006X
Elizabeth A VanSickle (ID) https://orcid.org/0000-0001-5504-9248

### Ethics

Human subjects: FDA approval (IND# 144022) was first acquired before therapy was initiated. The Spectrum Health IRB approved the study (IRB# 2019-161) and informed consent was acquired before blood sample collection. Consent was obtained for use of identifying patient images and videos within this manuscript. After study, authorization for publication was obtained from the parents of the child, now been placed in the medical records.

### Decision letter and Author response

Decision letter https://doi.org/10.7554/eLife.67097.sa1
Author response https://doi.org/10.7554/eLife.67097.sa2

# Additional files

## Supplementary files
- Source data 1. Global Metabolomics.
- Transparent reporting form

## Data availability
Data is provided in Source data 1.

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
