## [Decision Letter]

**Acceptance summary:**

The study elegantly demonstrates the use of an FDA-approved drug, Eflornithine, to treat, with considerable success, the neurological and skin manifestations of a child with a gain of function mutation of the ODC1 gene. The authors have addressed all of our concerns/comments and the manuscript in its revised form is now acceptable for publication at e*Life.***Decision letter after peer review:**

Thank you for submitting your article "Repurposing Eflornithine to Treat a Patient with a Rare ODC1 Gain of Function Variant Disease" for consideration by *eLife*. Your article has been reviewed by 2 peer reviewers, and the evaluation has been overseen by a Reviewing Editor and Mone Zaidi as the Senior Editor. The reviewers have opted to remain anonymous.

Essential revisions:

There was significant interest in the work overall as noted below from comments from both reviewers – the repurposing of FDA approved to treat existing diseases is of great interest to the scientific and medical community. There are however several issues that need to be addressed prior to the acceptance of this manuscript to e*Life*.

1) Where there side effects associated with administration of the drug and if so those be noted and described in detail with laboratory values/data provided when available

2) While the work is of interest a single case is presented, were additional patients treated and if so what was their outcome?

*Reviewer #1:*

In the present study, Rajasekaran et al. describe the clinical findings resulting from repurposing Eflornithine (ODC inhibitor), an FDA approved drug, for a metabolic rare disease caused by pathogenic gain-of-function variants in ODC1. Based on their previous case report (PMID: 30239107), the authors observed that the clinical findings of their patient (alopecia, brain and scalp cystic lesions, and a variety of metabolites) improved after Eflornithine treatment. The study provides intriguing new insights into the treatment of this ultra-rare condition, and highlights the success of repurposing drugs for metabolic disease, but is limited by the single patient treated and lack of information about side effects.

1. Eflornithine is an inhibitor of ODC. Were there any side effects observed in the patient during the treatment process, especially during the period of dosage adjustment?

2. Myelosuppression is common in patients taking Eflornithine. Were complete blood cell counts measured before, during, and after the treatment? These should be included in the report.

3. Did the protein level of ODC change before and after treatment?

4. In the Discussion section, the authors mention the possibility of replication of findings in the current study in other patients with ODC1 gain-of-function. There is another report of 4 individuals with ODC1 gain-of-function variants (PMID: 30475435) which highlights the variable expressivity present among patients with this disorder. Replicating their findings in some of these other patients seems necessary before drawing conclusions about the efficacy of Eflornithine for effectively treating the clinical features of ODC1 gain-of-function variants. This is especially important since the authors were unable to perform neurological assessments for their study.

*Reviewer #3:*

This is an excellent paper which describes a rational and effective treatment for a very rare inherited condition.

The authors explain clearly the reasons for their using the drug selected based on the available literature with a transgenic mouse model and show very encouraging preliminary results that it will be effective.

There are a few corrections/additions which should be made to the manuscript.

1. The sentence starting on line 58 is ambiguous. Are there for additional patients with similar mutations that have been reported and four more cases or is the total number of cases actually known at present.

2. Some details of the low polyamine diet that was co-administered with the drug should be given and some discussion of whether this is likely to be a critical issue in the success of the treatment are needed. Since, as described in the paper, polyamine levels are intricately controlled by regulation of synthesis, degradation and uptake, it seems unlikely that uptake dietary polyamines would be a major issue in this condition which involves high levels that would be expected to suppress uptake.

3. Figure 2C would be very confusing to anyone not familiar with the ODC structure. It needs a longer explanation in either the text or the figure legend explaining that the protein is made up of two identical subunits which form to active sites at their interface. Showing the two subunits in different colors would be helpful. it is also confusing to show the two ends in different formulations without clearly explaining why this is done. The text would better if it referred to the two ODC monomers than two ODC proteins

A reference to the structure is needed eg for human (Almrud, J. J., Oliveira, M. A., Kern, A. D., Grishin, N. V., Phillips, M. A., and Hackert, M. L. (2000) Crystal structure of human ornithine decarboxylase at 2.1 Å resolution: structural insights to antizyme binding. J. Mol. Biol. 295, 7-16 ) or mouse (Kern, A. D., Oliveira, M. A., Coffino, P., and Hackert, M. L. (1999) Structure of mammalian ornithine decarboxylase at 1.6 Å resolution: stereochemical implications of PLP-dependent amino acid decarboxylases. Structure 7, 567-581).

4. A reference to the original work of the very gifted chemists and pharmacologists that lead to the conceptualization and production of DFMO would be appropriate. For example, Metcalf, B. W., Bey, P., Danzin, C., Jung, M. J., Casara, P., and Vevert, J. P. (1978) Catalytic irreversible inhibition of mammalian ornithine decarboxylase (E. C. 4. 1. 1. 17) by substrate and product analogues. J. Am. Chem. Soc. 100, 2551-2553.

---

## [Author Response]

Essential revisions:There was significant interest in the work overall as noted below from comments from both reviewers – the repurposing of FDA approved to treat existing diseases is of great interest to the scientific and medical community. There are however several issues that need to be addressed prior to the acceptance of this manuscript to eLife.1) Were there side effects associated with administration of the drug and if so those be noted and described in detail with laboratory values/data provided when available

Additional information about the safety screening procedure for this treatment is now included. No abnormalities were noted from any of the laboratory values monitored, and there were no clinically apparent side effects observed throughout this treatment. An updated paragraph detailing this information is now included both in methods section and the patient’s experience in the result section of the manuscript.

Methods section “These draws were performed in conjunction with safety screening that included at baseline, weekly after each dose increase and then twice monthly, which included a complete blood count, liver function test, lactate dehydrogenase, complete metabolic panel, calcium, magnesium, and phosphate. Audiograms were obtained pretreatment and then monthly. Electrocardiograms were done pretreatment, 1 month after first dose, and every 6 months, all based upon guidance from previous pediatric treatment trials with DFMO^10^”

In the Results section “All pre-treatment testing results were within normal limits, and through the duration of treatment, no abnormalities were noted in any of the described safety monitoring. This includes no gastrointestinal adverse effects or myelosuppression, previously noted in patients treated with DFMO.”

2) While the work is of interest a single case is presented, were additional patients treated and if so what was their outcome?

No, as with most rare diseases, we have a unique situation but there are new patients being identified with potential for benefit from DFMO treatment. These patients are in other locations receiving healthcare from other Institutions. The process of initiating DFMO needs careful planning and FDA involvement. We hope to continue to understand this condition with the potential for expanded treatment, one day. The publication of this report will go a long way to shed light on this rare disorder and highlight potential for therapy. To the best of our knowledge, there are currently no other patients with this disorder that have been treated with DFMO.

Reviewer #1:In the present study, Rajasekaran et al. describe the clinical findings resulting from repurposing Eflornithine (ODC inhibitor), an FDA approved drug, for a metabolic rare disease caused by pathogenic gain-of-function variants in ODC1. Based on their previous case report (PMID: 30239107), the authors observed that the clinical findings of their patient (alopecia, brain and scalp cystic lesions, and a variety of metabolites) improved after Eflornithine treatment. The study provides intriguing new insights into the treatment of this ultra-rare condition, and highlights the success of repurposing drugs for metabolic disease, but is limited by the single patient treated and lack of information about side effects.1. Eflornithine is an inhibitor of ODC. Were there any side effects observed in the patient during the treatment process, especially during the period of dosage adjustment?

We thank reviewer 1 for identifying this shortcoming as did the prior reviewer. Please see our response to as we did not identify any adverse effects to the patient during this treatment. Statements to that are now in both the methods and Results section.

2. Myelosuppression is common in patients taking Eflornithine. Were complete blood cell counts measured before, during, and after the treatment? These should be included in the report.

Complete blood counts were part of the safety monitoring during this treatment, and we did have access to other historical CBC results for this patient. There was no observed difference in counts during treatment, and a specific comment has been added to the manuscript noting this as it is an important matter to comment on.

3. Did the protein level of ODC change before and after treatment?

This is an excellent question and this is a limitation. We had previously measured the ODC levels of this patient in red blood cells and primary dermal skin fibroblasts before treatment and its response to DFMO. The previous publications showed both ODC proteins and enzymatic activity were significantly elevated compared to healthy controls and the levels dropped after DFMO. This is consistent with the hypothesis that the C-terminally deleted ODC leads to accumulated and active ODC protein in the patient (due to failure of the proteasome to clear the protein), resembling a gain-of-function (GOF) mutation. These observations were published prior to this study (Bupp et al., AMJG, 2018; Schultz et al., Biochem J, 2019). The use of western blot to measure protein levels has remained the traditional pathway to detect levels of intermediates before designing therapies for rare diseases. In this case we used the information from the previous study supported with global metabolomics done at different timepoints during therapy to determine that Putrescine was indeed elevated and dropped on therapy. After discussions with the FDA we pursued therapy through the more rapid compassionate use mechanism. Our hope is that this manuscript will highlight this approach as a model for using repurposed drug therapy rapidly with rapid advances in genetic diagnostics and need for rare disease treatments.

A statement is now added to the discussion as a limitation that says: “A limitation of this study is that we did not specifically measure enzymatic levels. In rare diseases, the enzyme activity and the resultant metabolic perturbation are traditionally studied in bench settings by detecting biochemical changes in the patient’s biological samples. This study used global metabolomics to measure polyamine levels serially as the patient underwent therapy.”.

4. In the Discussion section, the authors mention the possibility of replication of findings in the current study in other patients with ODC1 gain-of-function. There is another report of 4 individuals with ODC1 gain-of-function variants (PMID: 30475435) which highlights the variable expressivity present among patients with this disorder. Replicating their findings in some of these other patients seems necessary before drawing conclusions about the efficacy of Eflornithine for effectively treating the clinical features of ODC1 gain-of-function variants. This is especially important since the authors were unable to perform neurological assessments for their study.

Indeed, the phenotype may be variable, and the exact metabolic derangements need careful examination before therapy is initiated. We knew from the cell studies that she exhibited gain of function dyshomeostasis. In addition we utilized global metabolomics to map out the derangements at multiple timepoints. The presence of CAP/CLIA mandates allows the results to be reported in the Electronic Medical Record and allows treating physician to indeed initiate treatments. Our hope is that this initial manuscript highlights an approach that is more clinically practical in laying the groundwork for additional study and serves as a model for other ‘n of 1’ treatments.

One of the main limitations in rare disease research is the recruitment of patients from all over the world even if several have been identified, the challenge becomes how to get patients’ local hospital systems interested in novel trials particularly when these are often non-academic settings and how to balance the costs to families if recruitment is out of network or requires extensive travel. Thus, our hope is that this paper could open the possibility of navigating these challenges. In addition global metabolomics can be used to characterize the dyshomeostasis in each of these patients in the original healthcare institution.

A statement has now been added to the discussion that says “There are challenges in treating patients with rare diseases as the exact phenotype may be quite variable and the exact metabolic profile needs to be first established. Once therapy is indicated then the healthcare institution needs to be willing to navigate the regulatory hurdles to make the specific agent available for a single patient**.”**

Reviewer #3:This is an excellent paper which describes a rational and effective treatment for a very rare inherited condition.The authors explain clearly the reasons for their using the drug selected based on the available literature with a transgenic mouse model and show very encouraging preliminary results that it will be effective.There are a few corrections/additions which should be made to the manuscript.1. The sentence starting on line 58 is ambiguous. Are there for additional patients with similar mutations that have been reported and four more cases or is the total number of cases actually known at present.

We agree that this sentence is poorly worded and confusing to the reader. There are four additional patients with similar mutations that were reported by Rodan et al. in 2018 as well as four more cases that have been identified but not yet reported. We will edit the sentence in line 58 to clarify this statement.

It now reads “Four additional patients with similar mutations and phenotypic features of this syndrome have since been reported^4^ with at least four more cases identified in other centers, but not yet reported”.

2. Some details of the low polyamine diet that was co-administered with the drug should be given and some discussion of whether this is likely to be a critical issue in the success of the treatment are needed. Since, as described in the paper, polyamine levels are intricately controlled by regulation of synthesis, degradation and uptake, it seems unlikely that uptake dietary polyamines would be a major issue in this condition which involves high levels that would be expected to suppress uptake.

The consideration of polyamines in diet, its impact with this syndrome, and the potential role during treatment is an excellent point. Considering this suggestion, we reviewed the diet of this patient further with her family. We provided them a list of foods that are high in Polyamines to avoid. The patient reported in this manuscript has significant developmental delay and receives most of her nutrition through formula (Pediasure) given through gastrostomy tube. Her diet remained unchanged during the study.

We added a sentence in the methods and in the discussion on our recommendation to the family to avoid dietary items that have a high Polyamine content. “The family was provided a list of high polyamine containing food to avoid and therapy with DFMO initiated”.

3. Figure 2C would be very confusing to anyone not familiar with the ODC structure. It needs a longer explanation in either the text or the figure legend explaining that the protein is made up of two identical subunits which form to active sites at their interface. Showing the two subunits in different colors would be helpful. it is also confusing to show the two ends in different formulations without clearly explaining why this is done. The text would better if it referred to the two ODC monomers than two ODC proteins

Thank You reviewer 3, This is now done. We have changed the text to refer to monomer units, changed one of the monomers to green, and added the following sentence to clarify the structure model interpretation, “The protein in figure 2C shows one of the monomers as a surface plot in gray and the second monomer in secondary structure in green, with the region of the patient removed colored in red.” We wish to point out to the reviewers that neither of the structures mentioned contain the C-terminal amino acids, and thus we did not use either of these structures alone. To get this model we utilized a nonbiased multi model homology strategy that filled in the end regions with loop samples. We have added into the methods the following sentence: “Protein model of ODC dimer was generated using a merge of PDB files 1D7K, 2OO0, and 2ON3 with samplings of other PDB files to fill in the C-terminal region, followed by relaxation of bonds with the YASARA2 force field in a water shell using YASARA tools (www.yasara.org/).”

A reference to the structure is needed eg for human (Almrud, J. J., Oliveira, M. A., Kern, A. D., Grishin, N. V., Phillips, M. A., and Hackert, M. L. (2000) Crystal structure of human ornithine decarboxylase at 2.1 Å resolution: structural insights to antizyme binding. J. Mol. Biol. 295, 7-16 ) or mouse (Kern, A. D., Oliveira, M. A., Coffino, P., and Hackert, M. L. (1999) Structure of mammalian ornithine decarboxylase at 1.6 Å resolution: stereochemical implications of PLP-dependent amino acid decarboxylases. Structure 7, 567-581).

Please see above, Thanks.

4. A reference to the original work of the very gifted chemists and pharmacologists that lead to the conceptualization and production of DFMO would be appropriate. For example, Metcalf, B. W., Bey, P., Danzin, C., Jung, M. J., Casara, P., and Vevert, J. P. (1978) Catalytic irreversible inhibition of mammalian ornithine decarboxylase (E. C. 4. 1. 1. 17) by substrate and product analogues. J. Am. Chem. Soc. 100, 2551-2553.

The reviewer makes an excellent point that the addition of a reference to recognize the research of those who worked to make treatment with DFMO possible in the first place would add great value to the manuscript. This is now ref 6 and a sentence recognizing this work and a new citation have been added. The sentence in the introduction reads as “Eflornithine is the result of work done on developing inhibitors that lead to irreversible enzymatic inactivation of ODC from nearly forty years ago.^6^”